# Flexible switch matrix addressable electrode arrays with organic electrochemical transistor and pn diode technology

Ilke Uguz [1] ✉, David Ohayon [2], Volkan Arslan[1], Rajendar Sheelamanthula[3], Sophie Griggs[4], Adel Hama [2], John William Stanton [1], Iain McCulloch [3,4], Sahika Inal [2] & Kenneth L. Shepard [1]

Due to their effective ionic-to-electronic signal conversion and mechanical flexibility, organic neural implants hold considerable promise for biocompatible neural interfaces. Current approaches are, however, primarily limited to passive electrodes due to a lack of circuit components to realize complex active circuits at the front-end. Here, we introduce a p-n organic electrochemical diode using complementary p- and n-type conducting polymer films embedded in a 15-μm -diameter vertical stack. Leveraging the efficient motion of encapsulated cations inside this polymer stack and the opposite doping mechanisms of the constituent polymers, we demonstrate high current rectification ratios ($10^5$) and fast switching speeds (230 μs). We integrate p-n organic electrochemical diodes with organic electrochemical transistors in the front-end pixel of a recording array. This configuration facilitates the access of organic electrochemical transistor output currents within a large network operating in the same electrolyte, while minimizing crosstalk from neighboring elements due to minimized reverse-biased leakage. Furthermore, we use these devices to fabricate time-division-multiplexed amplifier arrays. Lastly, we show that, when fabricated in a shank format, this technology enables the multiplexing of amplified local field potentials directly in the active recording pixel (26-μm diameter) in a minimally invasive form factor with shank cross-sectional dimensions of only $50{\times}8\ \mu m^2$.

Implantable electrophysiological neural interfaces couple electronics to neural tissue, representing a forefront research area for fundamental neuroscience and translational applications[1,2]. Most of the existing research in neural electrode technology has focused on passive electrode arrays that can provide a high-resolution interface for deciphering sensory input and enabling high-performance neuromotor prostheses[3]. Large-scale integration of such passive devices to match high-density neural populations is, however, challenging due to

the requirement to wire each electrode site to external electronics, leading to bulky interfaces[4,5]. A promising approach for reducing the area of the acquisition electronics is to support time-division multiplexing among multiple electrode sites after pre-amplification in the front-end electronics[6,7]. Successful implementations of this architecture rely on silicon-complementary metal-oxide-semiconductor (CMOS) technology with electrodes integrated directly on the CMOS substrates that are fashioned into a needle or shank form factor[8,9].

[1]Electrical Engineering Department, Columbia University, New York 10027 NY, USA. [2]Organic Bioelectronics Laboratory, Biological and Environmental Science and Engineering Division, King Abdullah University of Science and Technology (KAUST), Thuwal 23955-6900, Saudi Arabia. [3]Physical Science and Engineering Division, KAUST, Thuwal 23955-6900, Saudi Arabia. [4]Department of Chemistry, University of Oxford, Oxford OX1 3TA, UK. ✉e-mail: ilkeuguz@gmail.com

However, the mechanical mismatch between silicon-based electronics and soft tissue often results in an inflammatory response with glial scarring around the implanted electrodes[10,11]. Therefore, there is a critical need to develop soft, biocompatible electronics for neural applications to improve these mechanical interfaces with neural tissue[12,13].

To deliver soft active electronics, conducting polymer (CP)-based probes can be tailored to be highly flexible (with hardness below 100 MPa as compared to >100 GPa for even the thinnest silicon)[14,15]. Directly patterning CPs onto thin film plastics enables the creation of ultra-thin neural implants suitable for various electrophysiology applications, including recording and stimulation[16–19]. In particular, the integration of CPs in active devices leads to organic electrochemical transistor (OECT) technology, a high-gain transducer that uses aqueous electrolytes as the dielectric. In the OECT, upon application of a gate voltage ($V_{GS}$) between the electrolyte and the source of the OECT, the CP in the channel takes up electrolyte ions into its bulk, which translates into substantial changes in the source-drain current ($I_D$) generated by the application of a drain-source potential ($V_{DS}$) due to ion-coupled electronic charge compensation[20–22]. Current OECT-based active matrix addressers perform multiplexing by sequential application of $V_{DS}$ on single-element OECTs patterned in a network of scan and grid lines[23]. However, the high capacitance of CPs, which is essential for generating a high transconductance for the front-end amplifier, can lead to high leakage currents in the off state. Moreover, during active OECT switching, the charging currents at the channel/electrolyte interface can reach the range of micro-amperes. While these gate transients can be utilized for electroporation in cell cultures[24], they may lead to undesired neural activation in vivo[25]. In addition, as the OECT operation is conducted through a common electrolyte (i.e., cerebrospinal fluid) during in vivo recordings, these transients can lead to electronic crosstalk between adjacent channels. Therefore, to build large-scale electrochemical multiplexers, new strategies are required to select the active OECTs, such as integration with OFET switches[26]. Here, we overcome these limitations by multiplexing OECTs with an access diode in the form of a p−n organic electrochemical diode (p−n OED) that operates with internal ion motion and low leakage.

To accomplish this, we use complementary electron (n) and hole (p) transporting CPs and create a 15-μm-diameter p−n junction OED in a vertical cross-section. This device is enabled by the vertical-stack integration of a naphthalene-diimide-thiophene- (NDI-T)- based n-type semiconducting polymer bearing triethylene glycol (EG) side chains separated by a 6-alkyl carbon spacer from the NDI, referred to as p($C_6$NDI-T), and a p-type conducting polymer poly(3,4-ethylenedioxythiophene) polystyrene sulfonate (PEDOT:PSS). Metal contacts are patterned on the top (anode) and the bottom (cathode) of the stack. The doping state of each polymer film is altered by cation injection, but in opposite directions; that is, PEDOT:PSS undergoes oxidation while p($C_6$NDI-T) is reduced. After hydrating and charging these films with cations from the aqueous electrolyte, an anode-to-cathode bias ($V_{AC}$) reproducibly drives the encapsulated cations from one polymer to another. As described below, this internal cation motion within the volume of the polymer stack results in a rectifying behavior; that is, a depletion region is generated at reverse bias, and a high conductivity region is produced at forward bias, with this conductivity switch occurring within less than 250-μs. This model mimics conventional CMOS-based diodes, which rely on the generation of a depletion region in a p−n junction. However, in our case, the device utilizes ionic charge carriers.

This p−n OED model brings significant advantages compared to existing OED structures. The vertical configuration with encapsulated ions allows selective switching in a large matrix without any crosstalk with neighboring channels. In in vivo interfaces, where cerebrospinal fluid (CSF) covers all active sites, achieving such selectivity is not possible with planar configurations that rely on external electrolytes for charging[27]. Furthermore, our model stands apart from similar vertical OEDs that use hydrogels as ion reservoirs and CPs as diode channels[28]. In our p−n OEDs, CPs serve as both ion reservoirs and diode channels. As both polymers contribute to the output characteristics, the rectification ratio is significantly higher ($10^5 > 10^2$), which acts to reduce crosstalk.

We then form integrated circuits combining these electrolyte-compatible, ionic-charge-controlled, and fast-switching p−n OEDs with OECTs. We demonstrate that these OED−OECT circuits can overcome the limitations associated with the use of OECTs alone in the context of amplification combined with multiplexing. The internal ionic migration in p−n OEDs allows them to switch the current of on-state OECTs with negligible inter-channel crosstalk. We further generate exceptionally slender (8 μm × 50 μm), flexible neural implants using these OED−OECT circuits that deliver scalable channel counts for acquiring local field potentials (LFPs) with a signal-to-noise ratio (SNR) higher than 22 dB. These active multiplexers mark the first front-end implementation of fully organic electrochemical circuits for large-scale active neural recordings with more than 32 channels, doing so with a pitch of only 40 μm.

## Results
### p−n OED structure and characterization

An image of the p−n OED structure is shown in Fig. 1a. In this architecture; we utilize thin-film parylene C not only as a substrate (2 μm) and insulating layer (2 μm) but also as a vertical spacer (750 nm) to pattern the stacked anode and cathode Au contacts (Fig. 1b). The low dielectric constant of parylene C ($\kappa = 3.1$) ensures minimal crosstalk between metal interconnects, while the vertical stacking provides for a compact device footprint. Within the active 15 μm diameter circular pixel, we stack the two CPs of opposite transport polarity vertically between the anode and cathode (see Fig. 1c for the chemical structure of the polymers).

To characterize the CP films individually, we first pattern each polymer on a 20-μm-diameter Au electrode surface and measure their respective impedance spectra in buffer electrolyte (Supplementary Fig. 1). We find the double-layer capacitance of the n-type polymer p($C_6$NDI-T) to be 3.1 nF, approximately 1.2× higher than that of the p-type PEDOT:PSS film of the same thickness. For the maximum rectification ratio, only one of the CP films should contain the injected cations in the forward- or reverse-biased state. Therefore, we seek to balance the film capacitances in the p−n OED structure, which is achieved by choosing a film thickness of 200 nm and 240 nm for the n-type and p-type polymer films, respectively; leading to a capacitance of ~2.5 nF for both films in 15 μm diameter p−n OEDs. The polymer stack thickness increases by a factor of two over its dry state when hydrated and is kept hydrated to avoid the mechanical stress on the anode metallization that would be produced by repeated hydration/dehydration cycles. Wet tissue is, therefore, inherently advantageous for the operation of this technology that relies on constant hydration and ion motion. A patterned pinhole on top of the anode electrode (4-μm-diameter) is utilized for hydration and ionic charge injection.

Application of a positive voltage between an external Ag/AgCl electrode in the electrolyte solution and the bottom cathode electrode ($V_{EC}$) injects cations into the bulk of each polymer (Fig. 1b). Once the polymers are charged, we remove the electrolyte and disconnect the $V_{EC}$ circuit. We then apply a bias between the cathode and anode electrodes ($V_{AC}$), which causes the migration of the encapsulated ions between the stacked polymer layers in a direction determined by bias polarity (Fig. 1c). Application of a positive $V_{AC}$ (forward bias) drives the cations to the n-type polymer layer at the bottom of the stack, doping the film through cation compensation of the injected electrons and thus increasing its conductance. Unlike p($C_6$NDI-T), PEDOT:PSS is

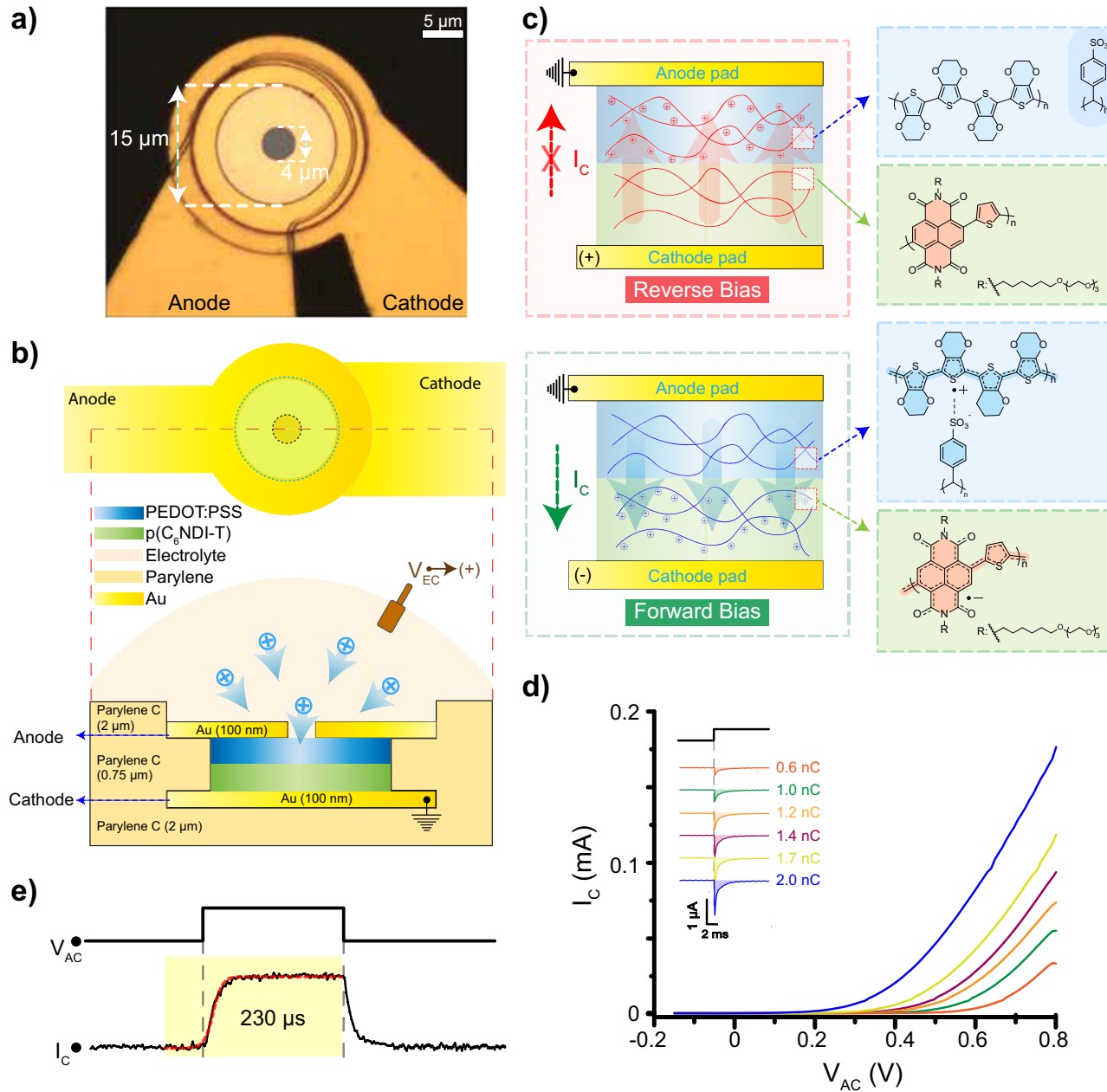

**Fig. 1 | Structure and performance of p–n OEDs with p–n junction. a** Optical image of the p–n organic electrochemical diodes (OED) as viewed from the top side. The area of the polymer coatings (highlighted in white) and the pinhole are marked by arrows with a diameter of 15 μm and 4 μm, respectively ($n = 20$ devices). **b** An illustration of the p–n OED from the top and side sections. The cations are injected inside the polymer films upon application of a bias between an external electrode and the bottom cathode contact. **c** Operating principle of fully charged p–n OEDs. Internal ion migration, based on the bias polarity between the top anode and bottom cathode electrodes, defines the on and off states of the device. Insets illustrate the electrochemical doping mechanism for each polymer film at reverse and forward bias with their molecular structures. **d** p–n OED performance as a

function of internal cation concentration. Current transients representing the charge injection inside the channel using an external Ag/AgCl electrode are illustrated in the inset of the figure. The injected charge in the films is noted next to each transient with values of 0.6, 1, 1.2, 1.4, 1.7, and 2 nC, respectively. $I–V$ curves of the p–n OED acquired as a function of injected charge. Every color corresponds to the $I–V$ response at various injected charge values noted in the inset in the same color. **e** Response time of the p–n OED corresponding to a 2 nC injected charge with a 0.8 V square pulse. ($n = 5$ repetitions) The $I_C$-time response is fitted with an exponential decay function (red dashed line). The saturation current is reached in 230 μs.

already conductive because the PEDOT chains are charge-compensated by the deprotonated sulfonic acid groups of PSS. As a result, at forward bias, the p–n stack is conducting. At negative $V_{AC}$ (reverse bias), on the other hand, the cations migrate to the bulk of the PEDOT:PSS film and compensate for the pendant sulfonate anions on the PSS. Holes are extracted at the anode pad, decreasing the hole density in the PEDOT film. Simultaneously, p($C_6$NDI-T) becomes depleted of cations, leading to high electrical resistance between

anodic and cathodic pads and giving the p–n OED unidirectional conductivity.

To investigate p–n OED performance, we apply various amplitudes of 10-ms $V_{EC}$ pulses (0.1, 0.2, 0.3, 0.4, 0.5, and 0.6 V) such that the polymer stack experiences different levels of cation injection. These levels of charge injection can be estimated from the double-layer charging current at the electrolyte–cathode interface (See Online Methods) to be 0.6, 1, 1.2, 1.4, 1.7, and 2 nC, respectively, as shown at

the inset of Fig. 1d. Depending on the number of cations encapsulated inside the bulk of the polymer, we observe significant changes in the resultant current–voltage (I–V) curves of the p–n OED, which we measure immediately after each charging pulse. The forward-bias electronic current level increases as a function of loaded cation concentration, resulting from the increased doping levels of the polymers (via an ion–electron charge compensation mechanism). At a $V_{AC}$ of 0.8 V, which we chose to remain within the water window, the stored charge in each polymer for a film capacitance of 2.5 nF is 2 nC. At this defined maximum injected charge level, the device has a rectification ratio of >10[5] as illustrated in Fig. 1d (see online Methods). Cyclic voltammetry (CV) curves for 15-μm-thick p(C₆NDI-T) films on Au electrodes show that above 0.2 V vs. Ag/AgCl, the p(C₆NDI-T) film turns fully conductive (Supplementary Fig. 2), which is consistent with this measured p–n OED turn-on voltage. This observation verifies that the device operation relies on cation migration in and out of the n-type film. In addition, since both the carrier concentration (upon doping) and electric field in the polymer stack increase linearly with $V_{AC}$, the forward-biased current characteristics show a quadratic dependence on $V_{AC}$. Overcharging the p–n OEDs to 4 nC results in a high off-state current in the range of -10 μA since the excess cations are trapped in the n-type polymer in reverse bias, leading to high leakage (Supplementary Fig. 3). Additionally, in this overcharged state in forward bias, the p–n OED current exhibits a linear dependence on $V_{AC}$.

The large-signal transient performance of the p–n OED is determined by the mobility of cations within the polymer stack. To measure this, we apply square wave $V_{AC}$ pulses switching between −0.8 V and 0.8 V. The corresponding chronoamperometry results show that the forward-bias DC level is reached in as fast as 230-μs with a symmetrical turn-off transient (Fig. 1e).

## Long-term stability of p–n OEDs

Despite the internal ionic conduction, cation diffusion out of the device through the top-contact anode pinholes represents a challenge for stable operation in aqueous environments. To restrict this out-diffusion, we seal the anode pinhole with a CP layer of undoped, p-type bithiophene–thienothiophene unit bearing ethylene glycol side chains, referred to as p(g2T-TT)[29]. To ensure better sealing and increase the length of the diffusion path out of the OED, primarily defined by the thickness of the p(g2T-TT) film, we further coat the top of the p(g2T-TT) seal with a thin film of SU-8 photoresist to define a planar length of 5 μm between the OED and the external electrolyte. The fabrication flow of this device architecture is illustrated in Supplementary Fig. 4.

Here, we leverage the low cation mobility inside p(g2T-TT) combined with the SU-8 capping layer to generate an ion barrier over the active p–n OED structure and encapsulate the cations inside the p–n OED. In CP films, the ionic charge drift is facilitated by water absorption[30]. For instance, PEDOT:PSS films absorb considerable water, swelling by over 100% of their dry thickness due to the polar PSS content. As a result, the cations in this film have high mobility, close to that of free ions in the electrolyte ($1.4 \times 10^{-3}$ cm² V⁻¹ s⁻¹ for K⁺)[31]. In contrast, despite the hydrophilicity induced by glycol side chains, p(g2T-TT) shows significantly less water absorption, swelling only ~10%. Therefore, ionic mobility in these films is significantly lower than in PEDOT:PSS[32]. Cations are also caged inside the glycolated side chains, further impeding their transport[33]. The primary reason to utilize this partial sealing here is to use the electro-active properties of the p(g2T-TT) films for sensing applications[29,32]. Note that permanent sealing with ion-selective membranes with high polycation content (e.g., polydiallyldimethylammonium chloride) could also be employed[34], and such strategies are commonly used in organic ion pump structures to achieve ion-selective transmission[35,36].

To characterize device stability, we compare the sealed p–n OED with an unsealed one (Fig. 2a). We operate both of these p–n OEDs as

half-wave rectifiers by applying a sinusoidal biphasic $V_{AC}$ with a peak-to-peak amplitude of 1 V at 20 Hz (Fig. 2b). For the unsealed p–n OED, direct exposure of the device to the electrolyte limits device operation to only a few cycles due to cation out-diffusion. Such "uncapped" architecture leads to an approximately three-fold decrease in the saturation peak current. In contrast, sealed devices do not show any immediate leakage. We further test the sealed p–n OEDs for over four hours under continuous operation. We do not observe any significant decay after the continuous application of 20,000 pulses (Fig. 2d). A peak current loss in the range of 10% is apparent after 30,000 pulses due to leakage through the polymer seal. This performance drop, however, can be compensated by re-injecting cations through reversible redox activity/charging of the CPs, ultimately enabling the OEDs to remain operational over hours (Fig. 2d).

## Active sensor integration with p–n OED in series

To use the p–n OED topology for electrophysiology applications, we integrate it with an OECT for the amplification of bio-potentials. Instead of using an external OECT, we utilize the p(g2T-TT) film that seals the p–n OED as an integrated OECT. This is achieved by positioning a source contact on the same layer as the anode contact of the p–n OED, which also functions as the drain contact for the OECT. We extend the p(g2T-TT) capping layer to the source pad as the exposed OECT channel (Fig. 3a). In this way, the distance between the source pad and SU-8 insulation overlaying the p–n OED defines the channel length of the OECT. This integration process offers a significant advantage over existing OEDs, as our p–n OEDs can be seamlessly integrated with OECTs using the same fabrication steps and temperature requirements. By employing this multi-layer fabrication protocol, the entire circuit can be integrated into a single pixel, which is comparable in size to a single neuron.

The p-type polymer we chose as the OECT channel (and capping layer), can be electrochemically doped with electrolyte anions (see Supplementary Fig. 5 for CV curves). Therefore, in the OECT configuration, a negative voltage between an external gate electrode and source contact ($V_{GS}$) injects electrolyte anions into the channel, doping p(g2T-TT). The doping levels are expected to be significantly lower under the SU-8 coating as the encapsulation over the drain contact limits the ionic drift underneath. Additionally, in the OECT channel, the electrochemical doping occurs in parts of the film that are closer to the source contact, where the voltage drop is the highest[37].

In the OED–OECT combination, the I–V characteristics of the series connection of both devices are defined by $V_{GS}$ and cathode-source potential ($V_{CS}$). During p–n OED switching, we keep $V_{GS}$ constant at a negative potential, maintaining the p(g2T-TT) film in a conductive state and simultaneously defining the anode potential. We switch on the OED–OECT by applying a negative $V_{CS}$. In this case, the p–n OED is conducting (on state), and the drain current of the OECT passes through the p–n OED and becomes the "cathode current" ($I_C$). Any modulation in the OECT current due to changes in the gate potential will thus be reflected in $I_C$. In this on-state, biopotentials in the electrolyte can be effectively amplified and recorded by the OED–OECT. At positive $V_{CS}$, the p–n OED is off, and there is no current flow in the OED–OECT device due to the rectifying action of the diode (off state).

For simultaneous operation of the bottom p–n OED and top OECT, there are some considerations on the geometry of the OECT channel. We first fabricate a single-element p(g2T-TT) OECT and perform electrical characterization by applying $V_{DS}$ and $V_{GS}$ biases. The p–n OED is characterized by a $V_{ON}$ of 0.18-V with an on-state conductance in the range of 250 μS (Fig. 1d). The on-state conductance of the OECT should be kept within this range to ensure high enough $V_{DS}$ during operation. To accomplish this, we scale the OECT channel length between the drain contact and SU-8 coating to be around 4 μm (Fig. 3a). Corresponding I–V characteristics of a single p(g-2T-TT) OECT with this channel length are shown in Fig. 3b. Biasing the $V_{DS}$ and

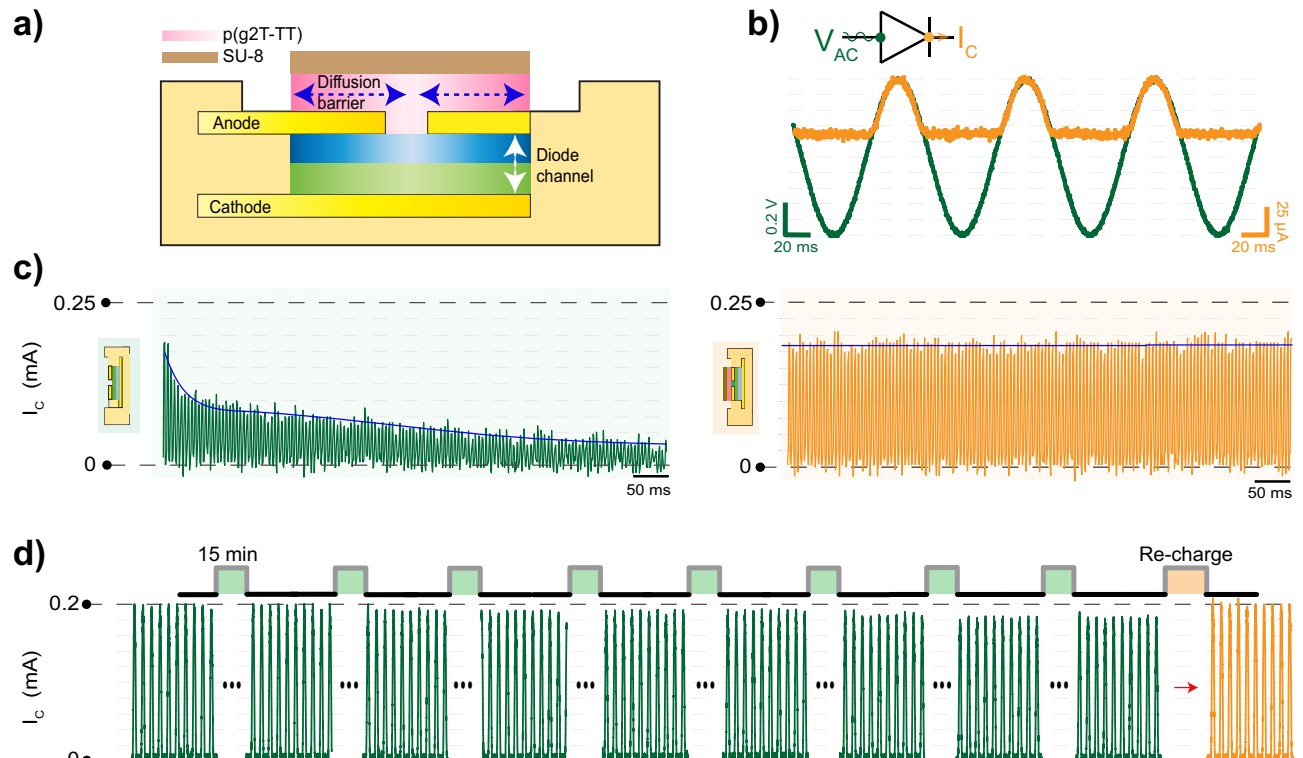

**Fig. 2 | Long-term stability of the p–n OED. a** An illustration of the p–n OED sealing with a top layer of p(g2T-TT) and SU-8. The blue arrow shows the diffusion barrier for cations between the OED and the overlying electrolyte. The white arrow indicates the p–n OED channel. **b** p–n OED operation as a half-wave rectifier. The green curve represents the sinusoidal input applied between the anode and cathode electrodes, and the orange curve is the output current of the p–n OED. ($n = 20$ repetitions per device. $n = 5$ devices.) **c** p–n OED operational stability performance without top sealing (in green) and with top sealing (in orange) during the application of 200-Hz cathodic pulses. The peak values of $I_C$ response is fitted with an exponential decay function (blue line). The corresponding device schematic is placed at the inset of each graph to illustrate the p–n OED architecture used. **d** Long-term performance of the sealed p–n OEDs during the application for 0.8-V biphasic pulses. Pulses shown in green (10 pulses at 10 Hz) are acquired after continuous application of 1000 pulses. Pulses shown in orange are acquired after p–n OED is recharged through a new cation injection.

$V_{GS}$ of the transistor at −0.3 V keeps the drain current ($I_D$) below 200 µA, which matches the OED current at a $V_{AC}$ of 0.8 V as shown in Fig. 1d. At the resulting $V_{CS}$ of −1.1 V, in-series operation of the OED−OECT structure has an on-state small-signal transconductance of over 1.5 mS. We also test the resulting rectification from the external gate electrode in the OED−OECT configuration. To do this, we operate both a single-element p(g2T-TT) OECT and an OED−OECT stack in diode configuration mode, where either $V_{DS}$ or $V_{CS}$ is directly connected to $V_{GS}$. In both cases, the output curves exhibit diode-like behavior with high conduction during forward bias. The leakage current, however, is three orders of magnitude lower for the OED−OECT stack due to the added resistance introduced by the OED when the stack is in reverse bias (Supplementary Fig. 6).

**Inter-channel crosstalk during OED switching**

To compare the level of crosstalk between OECTs in a large matrix experimentally, we pattern two neighboring OECTs and two neighboring OED−OECT stacks with a pitch of 40 µm, as shown in Fig. 3c, d. To consider crosstalk, we identify the channels being enabled as the aggressors and the adjacent unselected channels as the victims. We apply a constant $V_{GS}$ (0.3 V), generate square $V_{DS}$ pulses (switching from +0.3 V to −0.3 V) on the aggressor OECTs, and record the current waveforms in the adjacent victim OECTs. When the victim OECTs are in the off-state (with $V_{DS}$ at +0.3 V), we record a transient crosstalk in the range of ~µA. This results from the gate current of the aggressor OECTs, which partially flows to the drain of the victim OECTs due to the finite access resistance to the external gate electrode. Changes in the channel conductance of the victim OECTs due to the doping and de-doping

effects of this ion current are also observed, leading to steady-state crosstalk appearing as a shift in the baseline current with an amplitude of ~100 nA. Owing to the high resistance of the reverse-biased p–n OEDs, the crosstalk between an aggressor OED−OECTs (upon application of a constant $V_{GS}$ of −0.3 V and a square $V_{CS}$ pulse switching from −1.1 V to 0.3 V) and a victim OED−OECTs that are in the off-state (with a $V_{CS}$ of +0.3 V) is negligible. No crosstalk is observed at steady-state.

For an OED−OECT, the crosstalk in the victim channel from a single aggressor is more than 66 dB lower than the nominal on the current of 200 µA. Therefore, the total number of aggressors can be easily scaled into the thousands, enabling large-scale integration and addressability with minimal crosstalk. For single-element OECTs, because the crosstalk is 20 dB higher than the OED−OECT, the number of aggressors can only scale to the few tens before the crosstalk affects the baseline current.

Large ionic fluxes near active OECTs can also induce undesired neural activity. To test the level of this effect, we measure the gate current transients for both OECTs and OED−OECTs, as shown in Fig. 3e. We determine that the OECT gate transients lead to high charge injection per pulse (~10 nC), significantly higher than the neural stimulation threshold[25]. In the OED−OECT stack, however, the channel is in a doped state at all times, and gate current transients are more than an order of magnitude lower, which does not correspond to an injected charge level that can alter the spontaneous activity of nearby neurons.

**Active switch-matrix for time-division multiplexing**

We pattern our sealed stacked organic circuit between a grid of source and cathode electrodes, generating an 8×8 cross-point topology

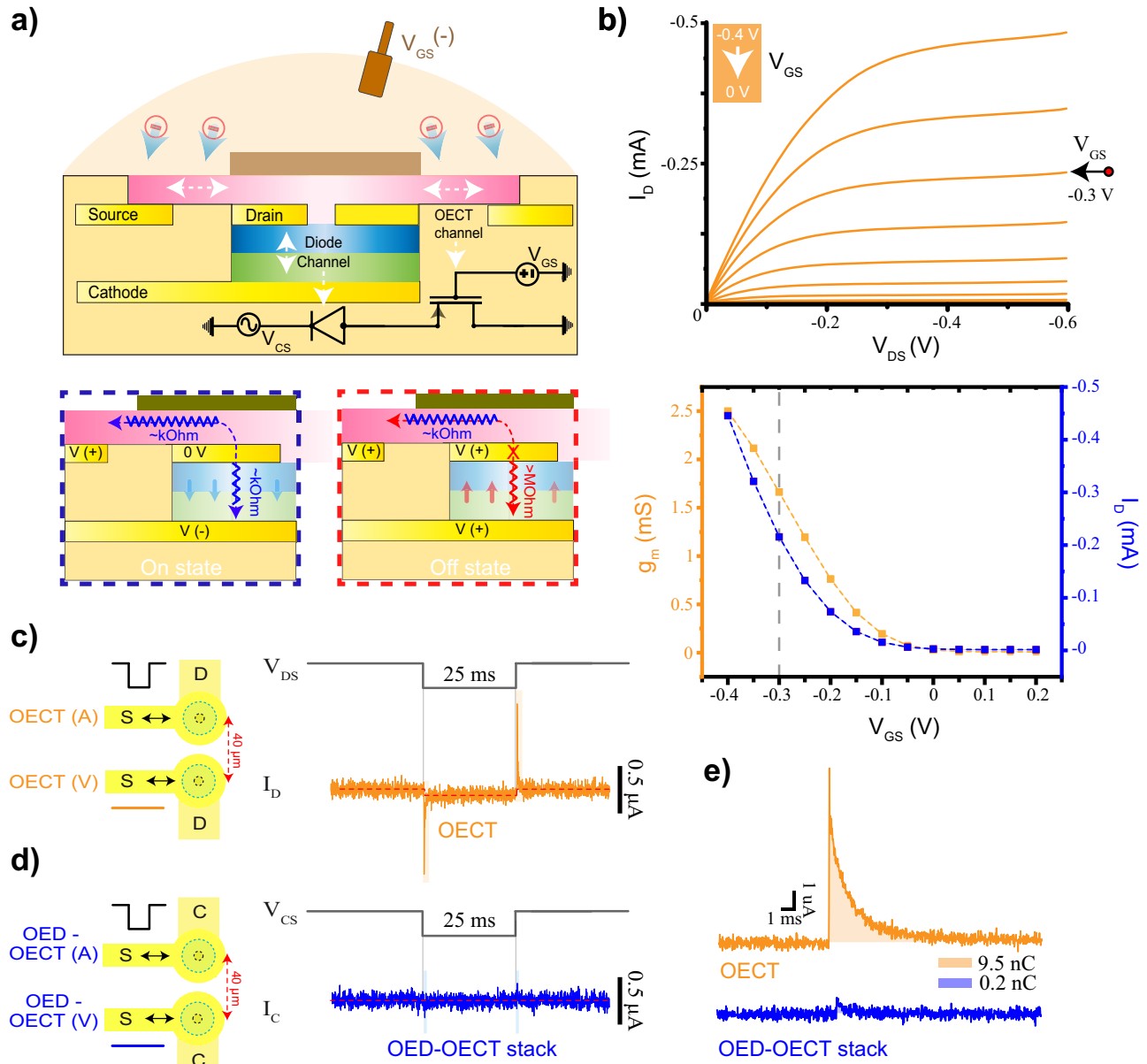

**Fig. 3 | OECT integration and inter-channel crosstalk. a** An illustration of the OED−OECT series configuration. An equivalent circuit is drawn under the device. Operation of the p(g-2T-TT) film in series with the p–n OED in the on and off states is illustrated in the bottom left and right insets, respectively. A pink color gradient is used to illustrate the estimated doping level of the p(g-2T-TT) film, where white represents low levels of doping. **b** I–V characteristics of a stand-alone p(g2T-TT) OECT. Output curves were measured for $V_{GS}$ values between −0.4 and 0 V with a step size of 0.05 V. The bottom plot shows the corresponding drain current (blue) and transconductance (orange) of the OECT for a constant $V_{DS}$ of −0.3 V ($n = 5$ devices.) The gray dotted line indicates the transconductance and the drain current values for $V_{GS} = −0.3$ V corresponding to the conditions used for in vivo experiments. **c** Crosstalk between OECTs. The left image illustrates two adjacent OECTs, which are patterned with a 40 μm pitch. Aggressor OECT (on the top) is used for

constant pulsing at 40 Hz, and victim OECT (on the bottom) is used for concurrent recordings. The top voltage profile represents cathodic pulses applied at aggressor OECT ($V_{DS}$). The bottom current trace represents the current response of the adjacent victim OECT ($I_D$). ($n = 5$ devices, $n = 3$ repetitions per device.) **d** Crosstalk between OED−OECT stacks. The left image illustrates two adjacent OED−OECT stacks, which are patterned with a 40 μm pitch. The aggressor OED−OECT (on the top) is used for constant pulsing at 40 Hz, and the victim OED−OECT (on the bottom) is used for concurrent recordings. The top voltage profile represents cathodic pulses applied at aggressor OED−OECT ($V_{CS}$). At the bottom is the current response of the adjacent victim OED−OECT ($I_C$). (n = 5 devices, n = 3 repetitions per device.) **e** Gate current transients of a single-element OECT (orange) and an OED−OECT stack (blue) during active switching. Corresponding charge injection levels, acquired from the integration of the gate current, are noted at the inset.

(Fig. 4a) with a 50-μm pitch. This circuit has 64 inputs and eight outputs and consists of eight eight-to-one multiplexers. Each column shares a common DC source, which is tied to the ground, and the current flowing through it contains the transduced signal. A constant $V_{GS}$ is applied across a single array. Applying a $V_{CS}$ on a specific cathode contact selectively biases the corresponding row and switches on the OECT−OED devices in each column at its cross-section.

Unselected columns present a resistance on the order of hundreds of MΩ because their corresponding p−n OEDs are reverse-biased, minimizing leakage.

We demonstrate the multiplexer by recording from four inputs of the eight-to-one multiplexer. We record from a single OECT channel (Column S1 in Fig. 4b) while turning on the four cathode contacts by applying −1.1 V $V_{CS}$ pulses to the associated rows (namely

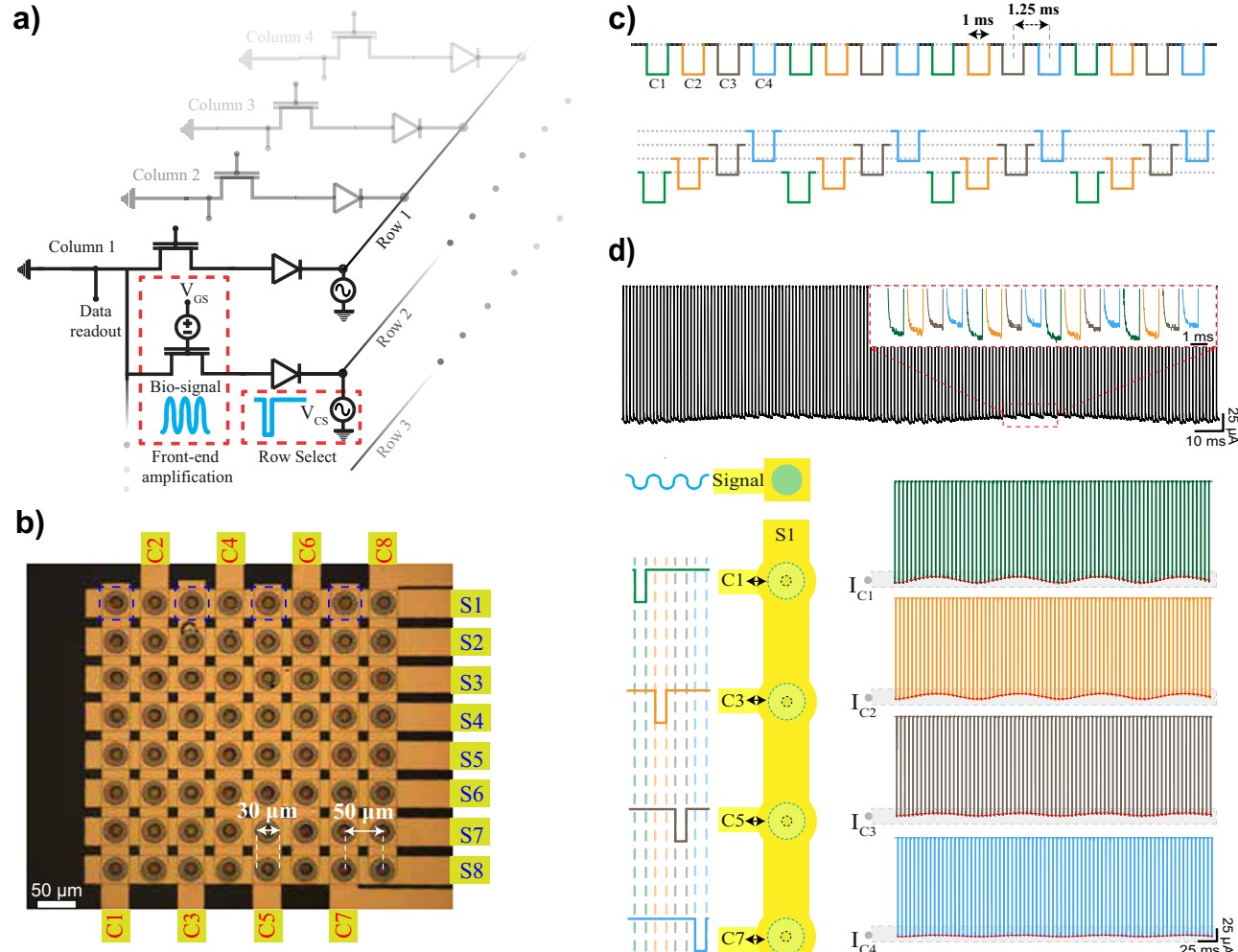

**Fig. 4 | Application of p–n OEDs for active multiplexing. a** Operation of p–n OEDs and transistors in series for multiplexing. A bias is sequentially applied between the source and drain (cathode) pads. Data readout is achieved by recording the OECT current. **b** A micrograph of the 8 × 8 cross-point topology. Eight sources and eight cathodes interconnects are patterned with a thin-film parylene-C spacer in-between. Polymers are patterned in a vertical form factor between gold layers ($n = 10$ devices). **c** An example of data readout acquired at S1 contact during the pulsing of drains at C1, C3, C5, and C7 contacts (highlighted by blue rectangles), which is achieved by enabling the underlying p–n OEDs. Cathode

pulses (0.3 V to −1.1 V) are applied relative to the source common with a 1 ms duration and 25% duty cycle with a period of 5 ms. **d** Operation of four inputs of the eight-to-one multiplexer in the electrolyte. A 50-Hz sinusoidal wave is generated with a sharp electrode located close to C1–S1. Sequential pulses (1 ms with 25% phase delay) are applied as in (**c**) and response is observed across all four channels. The top panel displays the unfiltered output acquired at S1, while the right panel illustrates the separated $I_C$ response as a function of channel position. The sections of the recordings that contain data are highlighted in gray for clarity.

C1, C3, C5, and C7) with a sampling rate of 200-Hz (See Online Methods). We then record the response from Column S1 and separate the four row-selected channels, as shown in Fig. 4c. Here, the front-end OECTs also act as anti-aliasing filters. We choose the p(g2T-TT) channel thickness to be over 250 nm, which leads to a −3 dB cutoff frequency of approximately 100-Hz (Supplementary Fig. 7). An external tungsten electrode (20-μm-diameter) placed near the C1–S1 stack is used to apply a 1-mV, 50-Hz sinusoidal voltage relative to the array source, comparable to the magnitude and frequency of LFPs as shown in Fig. 4d. The output current measured at S1 illustrates that the data exhibits variabilities depending on the specific switched OED–OECT stack in use. Consequently, separate waveforms are able to capture both the applied potential and its attenuation as a function of the distance from C1. The response acquired independently from all individual OED–OECT stacks during the application of external stimuli is demonstrated in Supplementary Fig. 8. These results, combined with the effective isolation of unselected channels, validate the utility of p–n OEDs in a switch-matrix configuration.

## LFP recordings with organic implantable multiplexer topology

We next use our conformable organic amplifier-multiplexer array to record LFPs in anesthetized mice. To accomplish this, we pattern the OED–OECT devices on a thin-film parylene substrate in a shank format. Each pixel is designed to have a 26-μm diameter, separated by an inter-transistor spacing of 20-μm. By vertically stacking four layers of metal interconnects, as shown in Fig. 5a, we pattern 12 metal interconnects on a 50-μm-wide shank to generate a 4 × 8 cross-point topology in which eight electrodes are multiplexed onto the reach of four output channels. The complete device thickness is only 8 μm and offers high mechanical flexibility, as illustrated in Fig. 5a, facilitated by the low Young's modulus of the parylene-C and nanofilm polymers constituting it (−0.1 to 2 GPa)[15,38,39]. We further show that p(g2T-TT) films that are exposed to the neural tissue in our multiplexer do not exhibit any toxicity in cell cultures (Supplementary Fig. 9, see online Methods).

To test the device in vivo, we open a 300-μm-diameter cranial window over the visual cortex (Fig. 5b). Exploiting the ultra-flexible nature of the parylene-based shank, we laminate it onto the opening with a micro-hole etched at the tip of the shank. By using a rigid glass

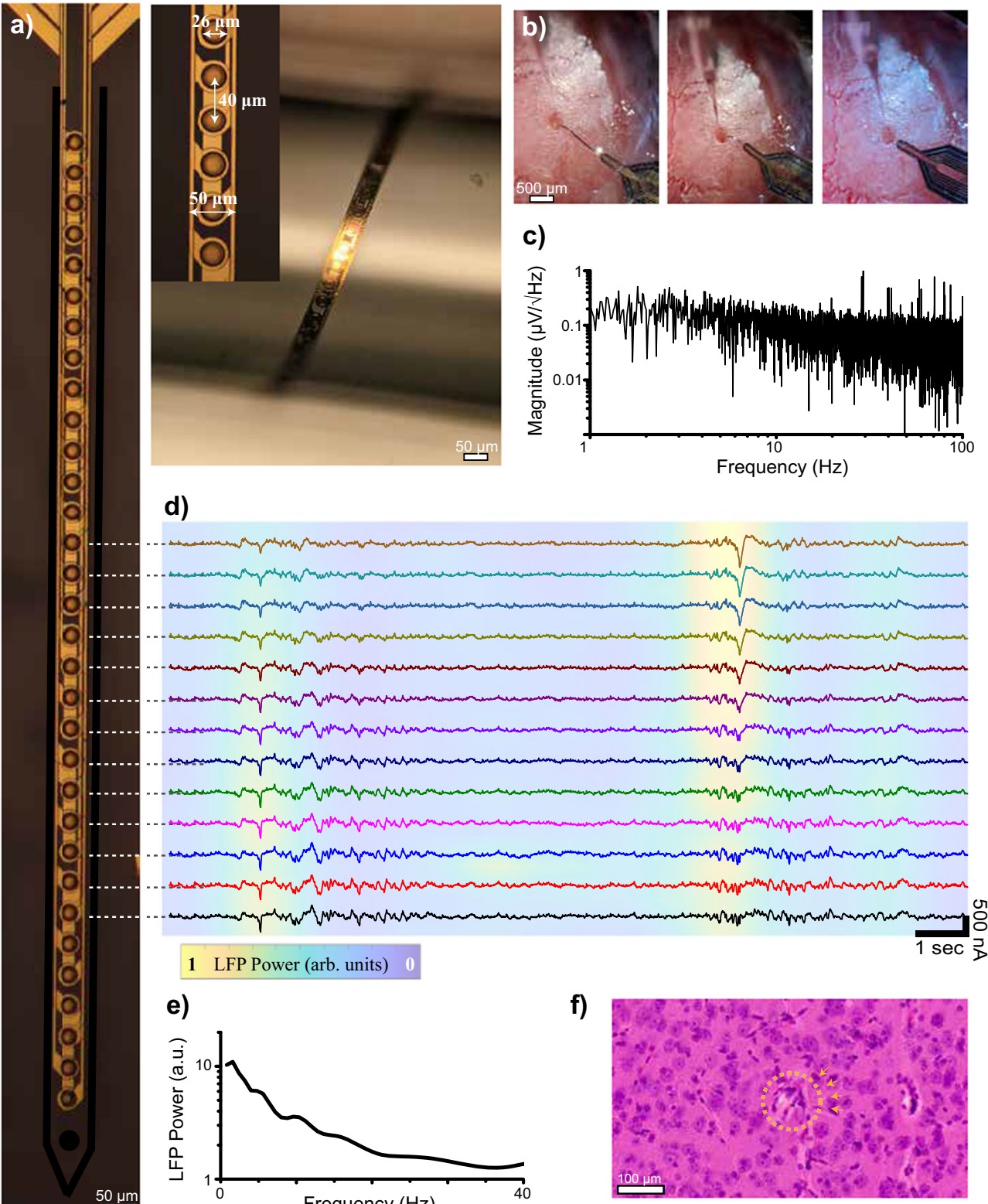

**Fig. 5 | Integration of 8 × 4 cross-point topography inside a 50 μm shank for in vivo LFP recordings. a** A micrograph of a 32-channel shank with 26 μm pixels and 40 μm pitch. In the inset, the flexibility of the shank is illustrated by wrapping the structure around a 1 mm pipette ($n = 8$ devices). **b** Implantation process of the multiplexer shank. The flexible shank is laminated on the exposed surface of the brain. A carrier glass pipette is anchored to the shank extension with a through-hole. A pipette is subsequently used to insert the shank and retracted to leave the fully flexible dual probe inserted inside the visual cortex of the brain. **c** Noise spectra of the OED–OECT stack acquired in saline within 0.1–100 Hz frequency range. **d** Local field potential waveform from the top 13 channels. The color map is expressed in arbitrary units as a function of LFP power within the 0.1–40 Hz frequency band and is added to illustrate the power decay of recorded LFPs as a function of distance ($n = 3$ implantations). **e** LFP power as a function of frequency within the 0.2–40 Hz frequency band. **f** H&E staining images were acquired 1 month after the probe implantation. The implantation zone is noted by an orange circle. The dashed orange circle delineates the implantation zone.

fiber guide, we insert the shank inside the tissue, driven by a micro-manipulator to a 1.5-mm depth. Retraction of the carrier shank leaves the flexible probe inserted in the brain with minimal tissue damage.

First, we measure the input-referred noise performance at one electrode, which we operate with only four inputs of the OECT–OED eight-to-one multiplexing configuration. By repeating patterns of 1.1 V-amplitude, 1-ms-wide pulses with a 5-ms period and a 25% phase delay between the pulses applied to adjacent cathodes, we can reach an effective sampling rate of 200-Hz per channel. The $I_C$ output is converted to a voltage by a custom transimpedance amplifier (TIA) board (Supplementary Fig. 10). Individual channel data is reconstructed from the raw multiplexed waveforms (see online Methods) and band-pass filtered to 100 Hz. Figure 5c shows the observed noise spectrum in the reconstructed channel recording; flicker noise dominates the 1–100 Hz range commonly associated with LFP recordings. In this case, the measured output $I_C$ is input-referred with a transconductance for the amplifying channel of approximately 1.5 mS. The noise spectra within the band from 0.15 to 100 Hz give a ~5.1 µV root-mean-square (rms) input-referred noise.

We then record LFPs along the shank; the top half of the shank (13 electrodes) covers the entire extent of Layers 1 to 6 in the mouse cortex. We apply four $V_{CS}$ pulses (to sequentially enable the top four columns in the array) and record the $I_C$ response from four separate sources simultaneously. Recorded channel activity after reconstruction is plotted in Fig. 5d. The spatial falloff of the recorded LFPs with cortical depth is clearly visible in the plots demonstrating the low crosstalk between channels. Pronounced spontaneous low-frequency oscillations (1–10 Hz) are characteristics of neural activity observed under anesthesia in mice (Fig. 5e)[40]. We compare the quality of OED–OECT stack recordings to those acquired by placing a 50-µm-diameter tungsten electrode next to the implantation site (Supplementary Fig. 11). The higher SNR of the organic stack (22 dB for the OED–OECT stack compared to 19 dB for the Tungsten electrode) validates the efficient front-end amplification of our multiplexer. We further show that the TIA does not provide additional amplification to the multiplexer output; instead, it results in a slight increase in noise (Supplementary Fig. 12 and see online Methods). Staining of ex vivo tissues suggests neither implantation damage nor scar tissue formation has occurred one month after implantation, apart from the minor injury created by the probe insertion (Fig. 5f).

## Discussion

Here, we realized a fully organic transistor-diode cross-point array structure with a 26-µm pixel and demonstrated its application for in vivo extracellular electrophysiology. The p–n OED, with its complementary materials, achieves rectification with encapsulated cations and allows individual addressing of OECT devices used for neural signal amplification. The diode cross-point configuration reduces reverse-biased leakage and eliminates the need for multiplexing through OECT gating, thus preventing crosstalk accros a shared CSF. The p–n OED structure, combined with the existing OECT technology, paves the way for CP-based large-scale active neural interfaces.

The device architecture presented here delivers 32-channel neural implants with a total volume of only 0.065 mm². Active multiplexing of the array allows this channel count to be supported with only 12 interconnects. Combined with the high mechanical flexibility of parylene as a substrate, our design represents an ideal model for large-scale integration of active electrodes with minimal disruption of tissue integrity. Moreover, the device structure demonstrated here is generated using basic cleanroom processing. Low-temperature and solution processing of the active materials allow for customizing the device performance and dimensions based on the target application. Here, we tuned the geometry of our devices to efficiently record the spatial distribution of LFP activity. In the future, these pixels can be realized on a much smaller scale to reach single neuronal resolution.

The performance of the p–n OED topology here is primarily defined by the p($C_6$NDI-T) film, which possesses both lower cation mobility and conductance compared to the p-type counterpart. Since the transient response of the p–n OED is dominated by the cation drift within the polymers and not by electron or hole transport, the low cation mobility of the n-type polymer limits the switching performance, and thus, the multiplexing rate. Advances in the field are required to give the n-type structures higher cation and electron mobilities. For instance, single-element internal gated OECTs have already been proven to work in the microsecond regime with p-type components[41,42]. The complementary operation of n-type polymers with similar performance will lead to faster switching characteristics for the associated p–n OEDs. Here, we tuned the geometry of the OECT to possess similar conductance values to the p–n OED. Improving the p–n OED conductance would allow shorter-channel-length OECTs to be used. Submicron channel lengths have already demonstrated $g_m$ values an order of magnitude higher than those in this study (~20 mS)[32,43].

Overall, the p–n OED topology presented here, as well as the front-end integration of n- and p-type polymers, are the first of their kind for neural interfacing. This work demonstrates that active organic electronic materials can form the basis of a high-density active front-end recording array, enabling the development of high-resolution neural interfaces with improved mechanical compliance.

## Methods

### Device fabrication

As the carrier substrate, a three-inch fused silica wafer was coated with a 2-µm-thick layer of parylene-C using a chemical vapor deposition (CVD) chamber (Specialty Coating Systems, PDS 2010). To pattern the first metal layer, a 1-µm-thick film of a lift-off resist (LOR3A) was first spin-coated and cured at 130 °C for 5 min. Subsequently, a 500-nm-thick film of an S1811 photoresist (Kayaku Advanced Materials) was spin-coated and cured at 110 °C for 1 min. The cured S1811 photoresist was then exposed to ultraviolet light (UV) using a Suss MA6 Mask Aligner to define the metal pattern and developed with AZ300MIF (MicroChemicals, Merck) developer for 25 seconds. An adhesion layer of 10-nm-thick titanium and a conduction layer of 100-nm-thick gold were sequentially deposited on the patterned photoresist by using an Angstrom EvoVac Multi-Process Evaporator. To complete the lift-off process, the metal coated wafers were soaked in Remover PG (Kayaku Advanced Materials, Inc., Westborough, MA) for 12 h. After completion of the first metal layer patterning, a secondary parylene-C film (0.75 µm) was deposited by CVD. To ensure strong adhesion, 3-(tri-methoxysilyl) propyl methacrylate (A-174 silane) was applied inside the chamber as an adhesion promoter before the deposition process. A second bilayer consisting of titanium and gold was deposited and patterned in a ring shape using parameters described before. Prior to coating with the top-layer insulation, a 1-µm-thick film of Shipley S1811 photoresist was patterned around the top circular electrode to protect it during the etching of top insulation. A third 2-µm-thick parylene-C film was deposited by CVD with an adhesion promoter. As an etch mask, an 8-µm-thick layer of AZ10XT (MicroChemicals, Merck) photoresist was spin-coated and cured at 115 °C for 2 min. The cured photoresist was exposed to UV light using a Suss MA6 Mask Aligner and developed with AZ400K (MicroChemicals, Merck) developer for 5 min. A dry-etch process was performed in a plasma-reactive ion etcher (Oxford Plasmalab 80; 180 W, 60 sccm $O_2$ and 2 sccm $SF_6$) until the top gold layer was reached. The residual AZ10XT and S1811 photoresists were rinsed off using Remover PG. As an anti-adhesion promoter, a soap solution diluted to 1% concentration in de-ionized water, was spin-coated over the wafer. A final layer of parylene-C was coated using CVD, which acts as a sacrificial layer to pattern the polymers inside the outer circle. An 8-µm-thick layer of AZ10XT was patterned using the parameters described above. A second dry etch process was

performed in a plasma-reactive ion etcher until the bottom gold layer and only within the area of the top circular electrode.

To prepare an aquous dispersion of PEDOT:PSS, PEDOT:PSS (Clevios PH1000) was purchased from Heraeus, whereas (3-glycidy-loxypropyl) trimethoxysilane (GOPS), 4-dodecyl benzene sulfonic acid (DBSA), 3-(trimethoxysilyl)propyl methacrylate (A-174 silane), ethylene glycol(EG) were purchased from Sigma-Aldrich. As channel material, PEDOT:PSS aqueous dispersions (95 wt%) were prepared and mixed with GOPS (1 wt%), DBSA (0.1 wt%), and EG (4.9 wt%). $p(C_6NDI$-T) syntheses was reported elsewhere[44]. $p(C_6NDI$-T) was first dissolved in chloroform (5 mg/mL) and spin-coated with a thickness of 200 nm. Then, the dispersion of PEDOT:PSS was spin coated with a thickness of 240 nm and baked for 1 min. To complete the patterning, the sacrificial layer of parylene-C was then peeled off. The wafer was baked at 110 °C for 15 min for PEDOT:PSS crosslinking. A final layer of gold with 300 nm thickness was deposited on top of the device. A 1-μm-thick film of Shipley S1811 was spin-coated, cured, and patterned to generate an etch mask to connect the top of the polymer with top circular gold electrodes with a micro-hole etched in the center. A wet etching step is performed using gold etchant for 150 s. The OEDs were immersed in de-ionized water for 12 h to release low molecular weight compounds and trapped ions. This completes the bare p−n OED fabrication. The p−n OEDs were charged with cations when the top metal electrode was still coated with S1811 in order to prevent cation accumulation at the anode electrode/electrolyte surface.

The residual photoresist was rinsed off using remover PG upon charging. For sealing, a top layer of p(g-2T-TT) was used, with syntheses reported elsewhere[32]. p(g-2T-TT), also dissolved in chloroform (5 mg/mL), is spin-coated over the OED. A layer of PMMA495, a protective layer to prevent exposure of p(g-2T-TT) to the developer in the subsequent photoresist development steps, was spin-coated with a 1-μm thickness and cured at 130 °C for 5 min. A 1-μm-thick film of Shipley S1811 photoresist is coated, cured, and exposed to act as an etch mask for p(g-2T-TT) patterning. A dry etch process is performed with a plasma-reactive ion-etcher as described above, and the residual photoresist is rinsed off using Remover PG. A 2-μm-thick layer of SU8 3005 was coated, cured, and developed in the center of the top p(g-2T-TT) channel. To generate shanks for in vivo experiments, the outline of the shanks was defined by using an IPG Photonics excimer laser.

## In vivo experiments

Columbia University's Institutional Animal Care and Use Committee (IACUC) conducted a thorough review and approval of protocols pertaining to the humane care and utilization of animals within the university's program. Additionally, they carried out inspections of both animal facilities and investigator laboratories. The assessment of the implanted devices was conducted in accordance with Animal Welfare regulations and Columbia University's IACUC guidelines (IACUC protocol number AC-AABE5554: "Development of high-density, implantable recording, imaging, and stimulating arrays"). For implantations, 6-month-old male wild-type mice (Strain:000664) were acquired from Jackson Laboratory. Housing conditions involved a 12-h light and 12-h dark cycle, with temperatures maintained in the range of 18 °C and humidity levels maintained in the 60% range. Mice were anesthetized using isoflurane in an induction chamber with an induction level of 3% at an $O_2$ flow rate of 2 L/min. After administering anesthesia, the mice were positioned on a stereotactic frame connected to a nose cone to ensure a continuous supply of isoflurane. To ensure the mice's body temperature was regulated throughout the surgery and recordings, the mice were kept on a feedback-regulated rodent warmer (ATC-2000) pad that was set at 32 °C. Prior to the incision of the scalp, 0.2 ml of bupivacaine was injected subcutaneously in the vicinity of the surgery site. Then the scalp over V1 was incised and the skull was drilled in a 0.5-mm-diameter circular shape. A borosilicate glass pipette with a 1.5-mm diameter was pulled using a Sutter Instrument Model P-97 with

an outer tip diameter of 10 μm. The pulled pipette was used as a rigid carrier to insert the flexible shank by anchoring it through a hole located at the tip of the shank. A second small opening was drilled over the cerebellum with a 0.5 mm-diameter circular shape, and a stainless-steel gate electrode was placed as the gate electrode. A dosage of 2 mg/kg of carprofen was administered via intraperitoneal injection into the animal's abdominal region.

## Histology

To evaluate the post-implantation tissue damage for acute damage (during insertion) and chronic damage (micro-motion of the brain), the flexible shank was implanted and kept in the brain for a month. Subsequently, mice were anesthetized using isoflurane as described above. Once anesthesia was induced, the mouse was decapitated, and the brain was removed. The extracted brain was then immersed in 10% formalin for a period of 72 h to complete the fixation process. The cortical tissues that were extracted were embedded in paraffin blocks and then horizontally sliced into sections. These sections were subsequently collected and prepared for hematoxylin-eosin (HE) staining. Images of the HE-stained tissue sections were captured using an Aperio LV1 slide scanner at a magnification of 40×. The entire process of sectioning the paraffin blocks, staining, and scanning was carried out by Columbia University's Molecular Pathology Core Facility (https://www.cancer.columbia.edu/research/shared-resources/molecular-pathology).

## Electrophysiology recordings and analysis

Sequential analog pulses for driving the multiplexer were generated by a LabVIEW-controlled data acquisition board (NI USB-6343). Extracellular electrophysiology data was coupled with an in-house printed circuit board with embedded TIAs for current-to-voltage conversion over a 250-Ω resistor. We record from a single OECT channel while turning on the four cathode contacts by applying 1.1-V amplitude $V_{CS}$ pulses to the associated rows with 1-ms pulse widths and a period of 5 ms with a phase delay of 25% between the pulses applied to adjacent rows. In this way, we achieve a sampling rate of 200 Hz. Raw multiplexed data is recorded by an oscilloscope (DSO-X 4034 A, Agilent Technologies). To reconstruct the original channel data, the multiplexed data was segregated using 1-ms on-state pulses specific to the chosen channel. Sample data points were taken from the saturation phase and each of these constituent channel waveforms was digitally filtered to a 100-Hz band.

## Noise tests

To measure the noise of the amplifier, we used a low noise resistor (Vishay, CH series) with 250 Ω resistance to simulate the OED−OECT stack. To precisely mimic the OED−OECT stack output current, we applied $V_D$ pulses with an offset of 50 mV, along with a 100-μV sinusoidal waveform. We subsequently measured the corresponding output voltage of the TIA using an Intan RHS2000 setup.

To evaluate the recording quality of our OED−OECT circuits, we conducted simultaneous recordings of neural signals using our OED−OECT stack and an external tungsten electrode connected to the Intan RHS2000 setup at a 30 kHz sampling rate. A high pass filtering stage (0.15 Hz) was integrated into the TIA board for compatibility with the Intan system.

## Biocompatibility studies

The cell line MCF 10 A (CRL-10317™) was provided from the American Type Culture Collection with lot number 70014922. The cell line is not in the list of known misidentified cell lines maintained by the International Cell Line Authentication Committee Authentication Committee. Human epithelial MCF10A cells were cultured using Mammary Epithelial Cell Growth Medium Basal Medium (MEGM, CC-3151, Lonza) supplemented with MEGM SingleQuots Supplement Pack (CC-4136, Lonza) without Gentamicin. 100 ng/mL Cholera toxin (Enzo Life Sciences), 100 U/mL Penicillin, and 100 μg/mL Streptomycin (Life

Technologies) were added to the media. MCF10A cells were placed and passaged in a humidified incubator at 37 °C, 5% $CO_2$.

5 μL of polymer p(g2T-TT) solution was spin-coated on a 12 mm diameter glass substrate. Glass-polymer-coated substrates were autoclaved at 121 °C for 15 min[45]. MCF10A cells were seeded on top of the polymer and incubated at 37 °C for 48 h. Media was then removed, and a mixture of Calcein-AM (Sigma), Propidium Iodide (Sigma), and Hoescht 33342 was applied to cells for 8 min at 37 °C. Subsequently, cells were washed twice with DPBS, calcium, and magnesium (Life Technologies) and imaged using an inverted fluorescent microscope.

An extraction method was used to assess the potential toxicity of the polymer. Polymer solutions were prepared as previously described and drop-casted onto glass substrates to maximize the amount of polymer in contact with the media. Coated substrates were then immersed in 700 μL of media (conditioned media) for 24 h at 37 °C, 5% $CO_2$. In the meantime, MCF10A cells were seeded in a 96-well plate to obtain quintuplicate per test. Then, an MTT assay (G4000, Promega) was performed according to the manufacturer's protocol. When cells reached adequate confluence (40 %), media was removed and replaced by fresh media (positive control), media with 0.01% Triton X-100 (negative control), and media conditioned (assay). Cells were then incubated in a humidified incubator at 37 °C, 5% $CO_2$ for 24 h. MTT dye was added to each well, and cells were incubated for 2 h at 37 °C, 5% $CO_2$. Then, after adding the dye dissolution solution, the absorbance was read at 570 nm against a reference absorbance at 650 nm using a microplate reader Infinite M1000 (Tecan).

### Device characterization

All $I-V$ characterization measurements are done using an electrolyte (PBS, phosphate buffered saline, 1× solution, Fisher BioReagents). $I-V$ curves are measured using a semiconductor device analyzer (B1500A, Agilent Technologies). An Ag/AgCl electrode (World Precision Instruments) is used as the external gating electrode. Response times and multiplexer measurements are performed using an oscilloscope (DSO-X 4034 A, Agilent Technologies). A 100-ohm resistor is connected between the drain output and the transistor or the p−n OED to achieve current-to-voltage conversion for time-response and noise measurements of the OED−OECT stack, independently of the TIA.

For p−n OEDs, we chose a minimum film thickness of 200 nm for p($C_6$NDI-T) since thinner films exhibited agglomerates and limited the p−n OED operation due to high reverse bias leakage currents. Chronoamperometry measurements using a semiconductor device analyzer were performed at various potentials to estimate the injected charge in the device stack. The injected charge at a given potential is obtained through the integration of the current value vs. time curve since the current flow between the electrolyte and polymer interface is primarily ionic. The forward-bias current at 0.8 V was recorded over 10,000 data points and averaged to calculate the current. The reverse-bias current at −0.8 V was recorded over 10,000 data points and averaged to calculate the off current. The ratio between the on and off currents was calculated as the rectification ratio.

Impedance spectra of the electrodes were recorded using an Autolab Potentiostat (Metrohm Autolab, Utrecht, Netherlands). A reference Ag/AgCl electrode and a counter platinum mesh electrode measuring 30 mm × 30 mm were placed inside the electrolyte. For impedance measurements, an AC voltage of 0.01-V amplitude was applied and the corresponding DC open-circuit potential against the reference electrode was acquired. For CV curve acquisition, the voltage ranged from −1 to 1 V, and the scan rate was 100 V/s with a step of 0.002 V. Multiple scan cycles were repeated for an accurate assessment of the shape of the voltammogram. For the undoped n-type polymer, impedance measurements were acquired at −0.4 V vs. reference electrode potential to ensure electrochemical doping of the measured film.

### Reporting summary

Further information on research design is available in the Nature Portfolio Reporting Summary linked to this article.

### Data availability

All data supporting the findings of this study are available within the article and its supplementary files. Any additional requests for information can be directed to, and will be fulfilled by, the corresponding author. The source data are available as a Source Data file for the respective figures. Source data are provided in this paper.

### Code availability

Custom Matlab code used in this study to analyze electrophysiology data is uploaded on github.com/klshepard/Electrophysiology-analysis-tools.

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

## Acknowledgements

This work was supported in part by the Defense Advanced Research Projects Agency (DARPA) under Contract N66001-17-C-4002 and by the National Institutes of Health under Grants U01NS099726 and U01NS099697 (K.L.S.). This work was performed in part at the Columbia Nano Initiative and part at the CUNY Advanced Science Research Center Nanofabrication Facility. This work is partially supported by King Abdullah University of Science and Technology Research Funding under Award No. ORA-2021-CRG10-4650 (S.I.).

## Author contributions

I.U. conceived the research. I.U. performed the neural implantations. I.U., D.O., and V.A. conducted the device characterization. I.U. and J.W.S. conducted the device fabrication. R.S. synthesized p(g2T-TT), and S.G. synthesized p(C6NDI-T), supervised by I.M. A.H. performed biocompatibility studies. I.U., D.O., S.I., and K.L.S. wrote the manuscript. K.L.S. provided overall project guidance.

## Competing interests

The authors declare no competing interests.
