## [Peer review file · Nature Communications]

REVIEWER COMMENTS

Reviewer #3 (Remarks to the Author):

The authors have appropriately addressed most of the reviewers' concerns and have taken measures to align the manuscript with a journal that is less focused on materials, allowing its contributions to be better appreciated by the intended readership. There are a few outstanding points that need to be addressed before I can recommend publication in Nature Communications:

1) The color map of the 8x8 array lacks sufficient information about addressability. Providing a comprehensive Fig. 4d in the SI that encompasses all channels, including the unselected ones, would make the findings more convincing.

2) The OECT shown in Fig. S6a is not configured as a diode, rendering the comparison meaningless. This issue still requires attention and resolution.

3) Firstly, the comparison of SNRs in Fig. S11 still spans two distinct domains (current vs. voltage). Secondly, the inclusion of the trans-impedance amplifier has introduced supplementary amplification and signal filtering (as indicated by the filter capacitors in Fig. S10). It is necessary to address this additional influence to establish a fair comparison.

Reviewer #3 (Remarks to the Author): *The authors have appropriately addressed most of the reviewers' concerns and have taken measures to align the manuscript with a journal that is less focused on materials, allowing its contributions to be better appreciated by the intended readership. There are a few outstanding points that need to be addressed before I can recommend publication in Nature Communications:*

Our response to Reviewer 3: We express our gratitude to the reviewer for dedicating their time and providing valuable input throughout the revision process, which has played a crucial role in enhancing the quality of our narrative. We trust that we have effectively addressed the outstanding concerns by conducting additional experiments, and we have elaborated on these experiments in detail below.

Reviewer 3 remark 1) *The color map of the 8x8 array lacks sufficient information about addressability. Providing a comprehensive Fig. 4d in the SI that encompasses all channels, including the unselected ones, would make the findings more convincing.*

Our response to remark 1) We have introduced more detailed experiments with a focus on the multiplexed architecture. Firstly, we have updated Figure 4 to include unfiltered output waveforms obtained from a single scan line. In this round of experiments, we have generated external signals using a tungsten microelectrode that matches neuronal dimensions (20 μm) and produces lower signal amplitudes (2.5-mV). For the initial test, we employed a 1x4 topology and presented the output signal of the multiplexer in Figure 4d. The output current, measured at a single scan line, demonstrates that the data exhibits variations depending on the addressed OED-OECT stack. These separate waveforms effectively illustrate both the applied potential and its attenuation relative to the distance from the stimulating electrode.

Additionally, in Supplementary Figure S8, we have addressed all the individual OED-OED stacks within an 8x8 matrix separately and plotted the responses obtained during the application of external pulses.

Line 302-311: “An external tungsten electrode (20 μm diameter) placed near C1-S1 stack is used to apply a 2.5-mV, 50-Hz sinusoidal voltage comparable to the magnitude and frequency of local field potentials (LFPs) relative to the source of the array (Fig. 4d). The output current measured at S1 illustrates that the data exhibits variabilities depending on the specific switched OED-OECT stack in use. Consequently, separate waveforms are able to depict both the applied potential and its attenuation as a function of the distance from C1. The response acquired independently from all individual OED-OECT stacks during application of an external stimuli is demonstrated in Fig. S8. These results, combined with the effective isolation of unselected channels, validate the utility of p-n OEDs in a switch-matrix configuration.”

Reviewer 3 remark 2) *2) The OECT shown in Fig. S6a is not configured as a diode, rendering the comparison meaningless. This issue still requires attention and resolution.*

Our response to remark 2) We now tested both OECT and OED-OECT architectures in diode configuration as suggested by the reviewer. For this purpose, we directly connected external V_{GS} electrodes either with V_{DS} pads for OECTs or V_{CS} pads for OED-OECT stacks. Upon application of a potential, both systems displayed diode like characteristics. Addition of the OED however, strongly decreased the reverse biased leakage current, effectively increasing the rectification ratio to 10^5 . This improvement indicates that the OED-OECT stack demonstrates a much better rectification performance compared to the OECT alone.

Line 239-244: “Furthermore, we test the impact of rectification the external gate electrode leads to in OED-OECT configuration. For this purpose, we operate both a single-element p(g2T-TT) OECT and an OED-OECT stack in diode configuration mode, where either V_{DS} or V_{CS} was directly connected with V_{GS} . In both cases the output curves exhibit diode-like behavior with high conduction during forward-bias. The leakage current however, is three orders of magnitude lower for OED-OECT stack due to added resistance introduced by the OED when the stack is at reverse bias (Fig. S6).”

Reviewer 3 remark 3) *Firstly, the comparison of SNRs in Fig. S11 still spans two distinct domains (current vs. voltage). Secondly, the inclusion of the trans-impedance amplifier has introduced supplementary amplification and signal filtering (as indicated by the filter capacitors in Fig. S10). It is necessary to address this additional influence to establish a fair comparison.*

Our response to remark 3) Since all the commercially available systems that we can reliably compare our array to are voltage-based, we decided to conduct the comparison based on the acquired signal-to-noise ratio. As suggested by the reviewer, it's true that amplification of the TIA and filtering stages can potentially affect this comparison. However, we carefully selected our filtering stages to operate within the frequency range of 0.15 Hz to 7 kHz. Our comparison was specifically conducted within the frequency band of 0.15 Hz to 200 Hz.

Additionally, we maintained the conversion resistor on the TIA at 250-ohms, ensuring that it primarily converted current without significant amplification. To validate this, we measured the noise and amplification of our system without the OED-OECT stack. To simulate OECTs, we utilized a low-noise resistor with a value of 250-ohms and applied a 0.05 V offset along its axis to generate a 200 μ A current, which is comparable to the output characteristics of OECTs. We also applied a sinusoidal waveform to V_D (100 μ V), and the resulting output waveform closely tracked the input waveform without any increase in the signal to noise ratio. On the contrary, there is a slight increase in the output noise of the TIA which can be attributed to the amplifier's inherent noise.

Line 349-350: “We further show that the TIA does not provide additional amplification to the multiplexer output; instead, it results in a slight increase in noise (Fig. S12 and see online methods).”

Line 510-514: “**Noise tests.** To measure the noise of the amplifier we used a low noise resistor (Vishay, CH series) with 250-ohm resistance to simulate the OED-OECT stack. To precisely mimic the OED-OECT stack output current, we applied V_D pulses with an offset of 50 mV, along with a 100 μ V sinusoidal waveform. We subsequently measured the corresponding output voltage of the TIA using Intan RHS2000 setup.”

REVIEWERS' COMMENTS

Reviewer #3 (Remarks to the Author):

The authors have successfully addressed the raised concerns. While questions still linger about the extent to which this technological advancement will revolutionize bioelectronics, the responsibility for assessing this lies within the community.